# Utility of Self-Reported Heat Stress Symptoms and NGAL Biomarker to Screen for Chronic Kidney Disease of Unknown Origin (CKDu) in Sri Lanka

**DOI:** 10.3390/ijerph181910498

**Published:** 2021-10-06

**Authors:** Pavithra N. Kulasooriya, Kithsiri B. Jayasekara, Thilini Nisansala, Sajani Kannangara, Ranawaka Karunarathna, Chaminda Karunarathne, Mahinda Wikramarathne, Steven M. Albert

**Affiliations:** 1University Hospital, Sir John Kotelawala Defence University, Colombo 10290, Sri Lanka; pavithra7kulasooriya@gmail.com (P.N.K.); tiliwijesinghe@gmail.com (T.N.); 2Faculty of Allied Health Sciences, University of Peradeniya, Kandy 20400, Sri Lanka; mwickramaratne@yahoo.com; 3Faculty of Allied Health Sciences, Sir John Kotelawala Defence University, Colombo 10290, Sri Lanka; kbjayasekara@kdu.ac.lk (K.B.J.); sajanikannangara123@gmail.com (S.K.); chamindasag@gmail.com (C.K.); 4Provincial Directors’ Office, Anuradhapura 5000, Sri Lanka; rhkarunarathna@gmail.com; 5Department of Behavioral and Community Health Sciences, University of Pittsburgh, Pittsburgh, PA 15621, USA

**Keywords:** chronic kidney disease, Sri Lanka, heat stress, agriculture, NGAL

## Abstract

Objective. We examined heat stress symptoms and urine markers of chronic kidney disease (CKDu) in Sri Lanka to assess differences between endemic vs. non-endemic regions and by occupation. Sample and Methods. We assessed a total of 475 villagers. In the endemic region, 293 were agricultural workers and 67 were not working primarily in agriculture. In the non-endemic region, 76 were agricultural workers. Of the residents, 218 were assessed for neutrophil gelatinase-associated lipocalin (NGAL), an early predictor of acute kidney injury, along with urine markers of chronic kidney disease. Results. The mean (sd) age of the sample was 45.2 (12.6), with males comprising 52.7%; 7.2% reported kidney disease (*n* = 34), and 5.7% reported diabetes (*n* = 27). The heat stress index (mean (sd)) was highest among agricultural workers in the endemic region (8.05 (5.9)), intermediate in non-agricultural workers in the endemic region (4.61 (4.5)), and lowest among agricultural workers in the non-endemic region (3.85 (3.3)); *p* < 0.0001. Correlations were higher between NGAL and serum microalbumin in the endemic agricultural worker sample than in the other two samples (Spearman’s r = 0.34 vs. 0.15 and 0.20). Conclusions. Both heat stress symptoms and NGAL values were higher among agricultural workers in endemic CKDu regions. Correlations between NGAL and microalbumin suggested a link between acute kidney injury and chronic kidney disease in the more-exposed sample.

## 1. Introduction

Climate change has led to increases in the prevalence of heat stress, including chronic dehydration, heat exhaustion, and heat stroke, with an increasing heat load on agriculture workers, especially in tropical regions [1]. These workers face increasing risk of chronic kidney disease in the absence of hypertension and diabetes (“chronic kidney disease of unexplained origin”, CKDu). CKDu has been reported in Asia (Sri Lanka, Bangladesh, and India), Africa (Egypt), and Mesoamerica (southern Mexico, Guatemala, El Salvador, Nicaragua, Honduras, and Costa Rica) [2,3]. Whether CKDu is one disease or varies by region and exposure remains controversial [4].

In Sri Lanka, heat stress may be associated with risk of chronic kidney disease among agricultural workers [5]. In prior research, we found greater heat stress and dehydration symptom burden among people in three villages from a high-prevalence CKDu region compared to people residing in a region with lower prevalence. Chronic dehydration and inadequate water consumption may thus increase the risk of kidney damage. These studies were carried out in Sri Lanka’s North Central Province (NCP), a thermally stressful area [6,7]. Temperatures range between 33.3 and 34.7 °C [8]. About 92% of the NCP population is engaged in agriculture [9]. If “heat stress nephropathy represent[s] one of the first epidemics due to global warming” [10], the NCP region of Sri Lanka will continue to face a growing CKDu disease burden, although the recent introduction of reverse-osmosis water processing in villages may have reduced CKDu incidence [11].

In this research, we combined samples from prior research [5] with a new series of participants to investigate heat stress symptoms in endemic vs. non-endemic regions, as well as variation by occupational type within endemic regions.

Our initial study did not find strong associations between standard urine biomarkers and kidney disease. Urine specific gravity, albumin, creatinine, and albumin–creatinine ratio were not strongly associated with reported heat stress symptoms. In our new sample from the same region, we added a new biomarker, neutrophil gelatinase-associated lipocalin (NGAL), an early predictor for acute kidney injury [12,13]. NGAL is primarily a marker of acute kidney injury. We reasoned that chronic kidney disease is likely preceded by an acute injury phase, and that NGAL levels may be higher in CKDu endemic regions. We also wished to see if combining the heat stress index and NGAL levels might be useful as screening tools for CKDu risk in the low-incidence population.

Our proxy for CKDu risk was residence in an endemic CKDu region, defined as a prevalence of 7% or greater based on health statistics from the Sri Lanka Divisional Secretariat. This epidemiologic approach allowed us to assess disease risk in the absence of serological markers of disease progression. The objective of the study was to examine acute kidney injury measured by NGAL values as a precursor to chronic kidney disease (CKDu) among agricultural workers in Sri Lanka, and to determine its relation to heat stress symptoms.

## 2. Methods

### 2.1. Participants

In 2019, we sampled 257 participants from villages in Vilachchiya and Nachchaduwa, the two main Divisional Secretariats (DS) of Anuradhapura District, North Central Province (NCP), Sri Lanka. As we reported, the prevalence of CKDu was high in Vilachchiya DS (7%) and comparatively low in Nachchaduwa DS (1.5%). In 2020, we sampled an additional 218 residents in the same region, and also in Hambanthota, Sooriyawewa. The sample villages varied in endemic status, and participants varied in occupational status. As shown in Table 1, we assessed a total of 475 villagers. We excluded pregnant women. In the endemic region, 293 were agricultural workers and 67 were not working primarily in agriculture. In the non-endemic region, 76 were agricultural workers. Study areas are in the dry zone of the country, with similar average temperatures of 32–35 °C, and similar agricultural cultivation. November to January is the rainy season in both endemic and non-endemic areas. The NCP Provincial Directors of Health reviewed the study protocol and approved the research.

Inclusion criteria for the agricultural worker samples were the following: manual work in land preparation, cultivation, and harvesting; outdoor exposure of at least 6 h per day, 4–5 days of the week; agriculture as major occupation for at least the last 5 years; frequent use of pesticides and agrochemicals; cultivation of 2+ hectares; age range of 25–55; and a resident of the endemic CKDu area for at least the last 5 years. Inclusion criteria for the non-agricultural sample included indoor work for at least 6 h per day, 4–5 days a week for the last five years; age range of 25–55; and a resident for at least 5 years in the endemic CKDu area.

### 2.2. Measures

As described in our earlier research [5], we adapted the US National Institute for Occupational Safety and Health (NIOSH) health hazard evaluation of heat stress (HETA-2012) [14], which we first translated and then back-translated into Sinhalese. Data included demographic information, history of agricultural and other work, secondary occupations, lifestyle-related information, medical history, and medication use. We collected additional information on land ownership, number of hours per day working in the field, resting hours, and water intake during working hours, both for agriculture and other types of work. Respondents reported the presence of diabetes mellitus, chronic kidney disease, hypertension, chronic back or joint pain, thyroid disease, muscle diseases (e.g., rhabdomyolysis), and other diseases diagnosed by medical officers.

The HETA questionnaire assesses the frequency (never, 1–2 days/week, or 3+ days/week) of 16 symptoms of potential heat stress and dehydration, including headache, “very dry mouth,” trouble urinating, fever, low urine output, exhaustion, nausea, muscle cramps, stomach/abdominal pain, dark urine, dizziness, heart racing or fluttering, diarrhea, disorientation, confusion, and vomiting. Reported frequencies were summed to construct a scale of potential heat stress–dehydration symptoms, with a range of 0–32. Psychometric properties of the Sinhalese HETA questionnaire were reported in prior research [5].

Research assistants from Sir John Kotelawala Defence University, Sri Lanka, carried out assessments under the direction of the principal investigator.

### 2.3. Laboratory Assessment of Urine Markers

Participants provided a urine sample that was analyzed in the laboratory. Participants were given a collection cup for 25 mL of urine, and samples were immediately frozen after centrifuging. Urine microalbumin was measured using turbidimetric immunoassay (BIOLABO, Lyon, France; detection limit 2–3000 mg/L) and urine creatinine using a modified Jaffé method. The albumin-to-creatinine ratio (ACR) was calculated. Urine specific gravity, an indicator of potential dehydration, was measured with a digital refractometer (PAL 10S; Serial No: P319450).

The laboratory protocol for NGAL is described in detail in the Appendix A. Briefly, urine NGAL was measured by the particle-enhanced turbidimetric immunoassay method using a Roche Cobas C 501 biochemistry analyzer. Only respondents from 2019–2020 completed this assessment (*n* = 218). Four samples were obtained over a 24 h period encompassing voiding in the morning, after work, in the evening, and in the morning on the next day. Urine samples were centrifuged at 3000 rpm for 10 min at stations set in the field directly after sample collection and within one hour. Not everyone completed all four samples, so in some cases analyses were limited to the first sample.

### 2.4. Analyses

We inspected means, variance, missing values, and potential outliers for all variables. Regression models were estimated to assess the association between the heat stress–dehydration index and risk factors for chronic kidney disease, including endemic region and occupation. Sub-analyses compared agricultural and non-agricultural workers in the endemic region. We used receiver operator characteristic curves to assess the sensitivity and specificity of the heat stress index for identifying residence in an endemic region. Spearman’s rank correlations were computed for urine biomarkers for each of the three samples. All analyses were conducted in STATA/SE 15.1.(StataCorp, College Station, TX, USA)

## 3. Results

The mean (sd) age of the sample was 45.2 (12.6). Males comprised 52.7% (*n* = 248); 7.2% reported kidney disease (*n* = 34), and 5.7% diabetes (*n* = 27). The mean (sd) for the heat stress index was 6.5 (5.5), and its range was 0–29 (logical range: 0–48). Missing data totaled <5% on all variables. Table 2 shows a comparison of the samples for key sociodemographic and clinical parameters.

### 3.1. Heat Stress Symptoms Were More Frequent in CKDu Endemic Regions

One-way analysis of variance showed that the heat stress index (mean (sd)) was highest among agricultural workers in the endemic region (8.05 (5.9), *n* = 288), intermediate in non-agricultural workers in the endemic region (4.61 (4.5), *n* = 67), and lowest among agricultural workers in the non-endemic region (3.85 (3.3), *n* = 115); F = 31.7, *p* < 0.0001, adjusted R^2^ = 0.18. Women reported a significantly greater frequency of symptoms than men (8.47 (6.0) vs. 4.79 (4.5), *p* < 0.0001). Two-way analysis of variance showed significant differences for gender (*p* < 0.0001) and endemicity region (*p* < 0.0001). The distribution of heat stress index scores for the three groups by gender is shown in the box plots in Figure 1.

Other univariate correlates of heat stress symptoms included self-reported diabetes (8.37 (5.5) vs. 6.4 (5.5), *p* < 0.04) and kidney disease (10.8 (7.1) vs. 6.2 (5.3), *p* < 0.0001). In a linear regression model (F = 18.5, *p* < 0.0001, adjusted R^2^ = 0.21) including age, BMI, gender, region/occupation (dummy coded), kidney disease, and diabetes, significant correlates included gender (b = 3.0, higher for women, *p* < 0.0001), kidney disease (b = 3.6, *p* < 0.0001), and endemic vs. non-endemic region (b = 2.7, *p* < 0.0001).

To examine differences by endemic region more carefully, analyses were restricted to agricultural workers in the endemic and non-endemic regions to compare heat stress symptoms. Figure 2 shows the distribution of the heat stress symptom index in the two groups. The mean for agricultural workers was 4.2 points higher in the high CKDu prevalence region (95% CI, 3.0–5.4, *p* < 0.001). Notably, all people reporting kidney disease resided in the high CKDu prevalence region.

The heat stress symptom index effectively identified the endemic region among agricultural workers. The area under the curve (AUC) defined by heat stress symptoms (classifier) and endemic region (outcome) was 0.727 (95% confidence interval: 0.68–0.79) using a non-parametric estimation with 1000 bootstrapped samples. The receiver operator characteristic curve (ROC) is shown in Figure 3. The AUC is interpreted as the probability that a randomly selected member of the endemic sample would have a higher heat stress symptom score than a randomly selected member of the non-endemic region population. A score ≥5 had a sensitivity of 0.64 and specificity of 0.64 for correctly identifying the endemic region.

### 3.2. NGAL and CKDu Risk

In the NGAL biomarker subsample, we processed at least one NGAL sample for 75 agricultural workers in endemic region, 67 non-agricultural workers in the endemic region, and 76 agricultural workers in the non-endemic region. The mean (sd) age was 42.4 (11.2), 69.2% were male, 5.2% reported diabetes, and 2.4% reported kidney disease.

NGAL-A (mean (SE)) values were higher in the endemic region (agricultural: 52.9 (24.5), non-agricultural: 48.9 (13.3)) than in the non-endemic region (agricultural: 17.5 (2.0)), but differences were not significant due to the high variance of NGAL values and small samples. Using all four NGAL samples (A–D) for the outdoor agricultural workers and tracking the number with any value ≥100, we saw a higher proportion in the endemic region (12.0%) than in the non-endemic (7.9%), but again, differences were not large enough to be significant.

Spearman’s rank correlations suggested a stronger relationship between NGAL and urine microalbumin in the endemic agricultural worker sample than in the other two samples. The correlation between NGAL and urine microalbumin was 0.34 (*p* = 0.02) in this subsample but was lower in the other two subsamples (0.15 in the endemic non-agricultural worker sample (*p* = 0.24) and 0.20 in the non-endemic agricultural worker sample (*p* = 0.08)). In all three groups, the correlation between NGAL and creatinine was similar (0.25, 0.27, 0.22).

In addition, all four time-sampling measures (A–D; available only for the two samples of agricultural workers) showed large differences between the endemic and non-endemic regions, as shown in Figure 4. Again, none of the means were significantly different due to the large variance of NGAL measures in the endemic region. For people reporting kidney disease (*n* = 5 in the sample), by contrast, all NGAL values were significantly elevated compared to people not reporting disease (samples A–D: kidney disease, 119.2, 160.0, 110.5, and 426.0; no kidney disease, 30.8, 26.1, 35.0, and 19.4).

In the NGAL sub-sample, heat stress symptoms (mean (SE)) were highest among agricultural workers in the endemic area (5.4 (0.50)), intermediate among non-agricultural workers in the endemic region (4.6 (0.55)), and lowest in the non-endemic region (2.8 (0.27)). In an adjusted linear regression model (Table 3), NGAL-A was significant as an independent correlate (*p* = 0.012) of heat stress symptoms; but regression diagnostics showed that the relationship was heavily dependent on a small number of people with large values on both measures. Endemic region, gender, diabetes, and NGAL were all significant independent correlates of heat stress symptoms.

As a final assessment of the relationship between heat stress symptoms, endemic region, and the NGAL biomarker, we defined four groups, cross-classifying endemic region (yes or no) with a high heat stress symptom score (<5 or ≥5) based on our classificatory model. Twenty-nine people were in the highest-risk category. Their NGAL-A mean was three times higher than people living in non-endemic areas who also reported a low heat stress symptom burden (93.0, *n* = 29 vs. 29.4, *n* = 107).

## 4. Discussion

This research showed that both heat stress symptoms and NGAL values were higher among agricultural workers in endemic CKDu regions. Heat stress symptoms and NGAL values were highest in people reporting kidney disease, all of whom resided in the endemic CKDu region. The heat stress symptom index reliably distinguished people in endemic and non-endemic regions, though these results must remain tentative without a biomarker of CKDu risk. Combining the self-report index with NGAL biomarker assessment may be useful for population screening.

A strength of this research was its purposive sampling of villagers in endemic vs. non-endemic regions and further separation of the endemic sample according to agricultural work. In the endemic region, heat stress symptoms were highest among agricultural workers, followed by non-agricultural workers from the same region, and followed finally by agricultural workers in the non-endemic region. This gradient was plausible, since non-agricultural workers in the endemic region may still do some farming, drink the same water, and face greater temperatures [5].

Further work must be carried out to assess the utility of NGAL for identifying incipient CKDu. Notably, variance in the measure was much greater among residents in the endemic region relative to the non-endemic region, as shown in Figure 4. This suggested that other factors were likely involved. Other biomarkers should be investigated. We stress that these findings will help mostly in identifying cases and will be most useful for screening. The mechanism of CKDu remains unclear. The strong association between endemic region and heat stress, now replicated in this follow-up investigation, showed that high temperatures and agricultural work are risk factors for CKDu. We are still missing the mechanisms that link region and occupation to increased CKDu risk.

To investigate potential mechanisms, we assessed the relationship between NGAL, a marker of acute kidney injury (AKI), and urine measures of chronic kidney disease (CKD), including microalbumin and creatinine. We found that correlations were highest in the endemic agricultural worker sample, suggesting a link between acute kidney injury and chronic kidney disease in the more exposed sample. AKI and CKD are very common noncommunicable diseases and represent a spectrum or continuum of kidney disease. Recurrent AKI is a causative factor for chronic kidney disease [15,16]. Non-recovery from repeated AKI episodes leads to persistent CKD, a more rapid transition between stages of CKD, and increased risk for progression to end-stage renal disease (ESRD) and the need for renal replacement therapy [17]. Therefore, early diagnosis and initiating treatment for AKI is important. However, it may be difficult to identify AKI, as symptoms may not be obvious to patients or specific to the disease [18]. Tests that help to rule out AKI and that use biomarkers such as NGAL may aid in diagnosis and assessment of AKI risk.

One screening recommendation based on this research is the potential value of the heat stress index for CKDu risk. It would be valuable to know if a greater heat stress symptom burden among agricultural workers in non-endemic areas, or non-agricultural workers in endemic areas, is a risk factor for CKDu. A longitudinal observational study tracking these cohorts would be required to assess risk differences, which may help in identifying the etiology of CKDu. Serial assessment of NGAL also may help in this effort.

## Figures and Tables

**Figure 1 ijerph-18-10498-f001:**
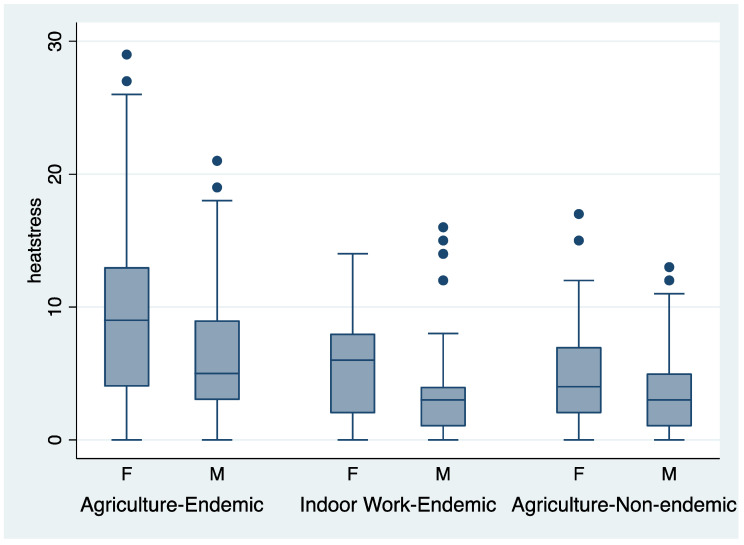
Box plots of the heat stress symptom index by region and gender. •- dots indicate extreme observations (>1.5 * interquartile range).

**Figure 2 ijerph-18-10498-f002:**
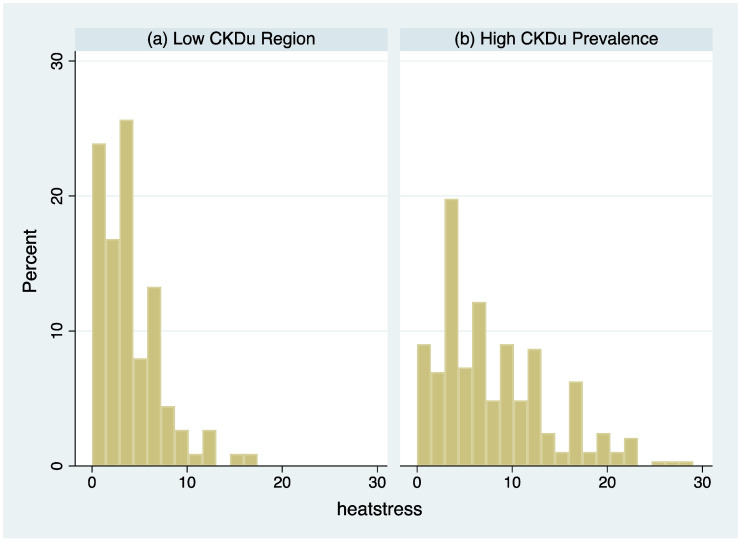
Histogram of the heat stress symptom index among agricultural workers: (**a**) Low CKDu Region, (**b**) High CKDu Region.

**Figure 3 ijerph-18-10498-f003:**
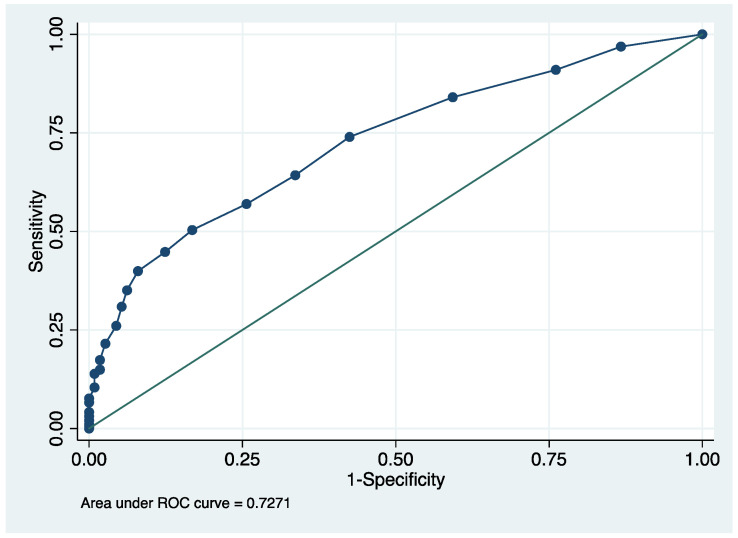
The receiver operator characteristic curve by heat stress and endemic region for agricultural workers.

**Figure 4 ijerph-18-10498-f004:**
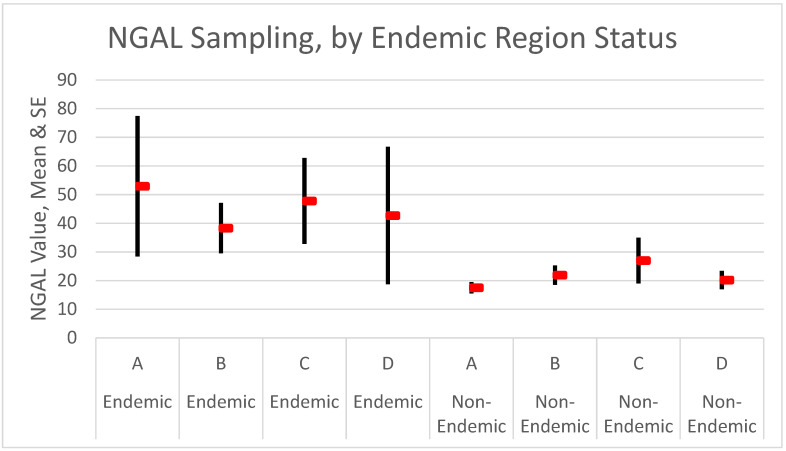
NGAL by endemic region. Values for the endemic region (**A**–**D**) were 72, 69, 65, and 68; and for the non-endemic region were 76, 74, 71, and 70. Red bars indicate mean values and the lines standard errors.

**Table 1 ijerph-18-10498-t001:** Distribution of Sample, by Region and Occupation.

Region	Date	CKDu Region and Occupation (*n*)	Villages (*n*)	Biomarkers
Anuradhapura District, Thanthirimale	May 2017	Endemic Agricultural (190) Non-agricultural (28)	3	Urine SG Urine albumin Urine creatinine ACR
		Non-endemic Agricultural (39)	1	
Anuradhapura, Thanthirimale	May–August 2019	Endemic Agricultural (75) Non-agricultural (67)	3	Urine SG Urine albumin Urine creatinine ACR Urine NGAL
Hambanthota, Sooriyawewa	March 2020	Non-endemic Agricultural (76)	3	Urine SG Urine albumin Urine creatinine ACR Urine NGAL

**Table 2 ijerph-18-10498-t002:** Total village sample and NGAL sub-sample.

**Total Village Sample**				
**Region**	Endemic Region		Non-Endemic	
**Occupation**	Agricultural Workers (*n* = 288)	Non-Agricultural Workers (*n* = 67)	Agricultural Workers (*n* = 115)	*p*
Demography				
Female, %	57.1	43.3	25.2	<0.001
Age	46.6	43.4	42.9	0.001
Health Conditions				
CKD, %	11.1	0.0	0.0	<0.001
Diabetes, %	5.2	10.5	3.5	NS
BMI	23.4	23.4	22.2	0.014
Heat stress	8.0	4.6	3.8	<0.001
Water Intake				
l/day	2.7	2.1	2.1	<0.001
**NGAL Sub-Sample**				
**Region**	Endemic Region		Non-Endemic	
**Occupation**	Agricultural Workers (*n* = 71)	Non-Agricultural Workers (*n* = 67)	Agricultural Workers (*n* = 76)	*p*
Demography				
Female, %	21.1	43.3	29.0	0.017
Age	42.8	43.4	40.5	NS
Health Conditions				
CKD, %	7.1	0.0	0.0	0.005
Diabetes, %	4.3	10.5	1.3	0.044
BMI	21.9	23.4	22.0	NS
Heat stress	5.4	4.6	2.8	<0.001
Water Intake				
l/day	2.0	2.1	1.7	<0.001
Urine Parameters				
Specific gravity	1.015	1.012	1.016	0.004
NGAL (se)	52.9 (24.5)	48.9 (13.3)	17.5 (2.0)	NS
Microalbumin (se)	17.9 (5.4)	13.5 (2.8)	25.8 (6.1)	NS
Creatinine (se)	158.4 (14.3)	107.0 (8.7)	130.7 (8.2)	0.007
ACR (se)	26.9 (11.6)	18.8 (3.9)	26.6 (8.0)	NS

Note: one-way analysis of variance was used. NS, not significant; SE, standard error.

**Table 3 ijerph-18-10498-t003:** Correlates of heat stress: regression model and NGAL sample.

Correlate	b	se	t	*p*	95% CI
Constant	6.89	2.86	2.41	0.02	1.26, 12.5
NGAL (morning sample)	0.005	0.002	2.54	0.012	0.001, 0.008
Age	0.004	0.023	0.18	0.86	−0.04, 0.05
Gender (male)	−1.61	0.57	2.84	0.005	−2.73, 0.49
Kidney disease	−1.67	1.71	0.97	0.33	−5.05, 1.71
Diabetes	2.83	1.22	2.33	0.02	0.43, 5.24
BMI	−0.01	0.06	0.16	0.87	−0.13, 0.11
Sample					
Agriculture–Endemic	Ref				
Indoor–Endemic	−1.40	0.66	2.12	0.04	−2.70, −0.10
Agriculture–Non-endemic	−2.62	0.63	4.15	<0.001	−3.87, −1.37

Note: *n* = 207, F (8, 198) = 5.37, *p* < 0.001, adjusted R^2^ = 0.15. Ref, reference group for dummy-coded indicator variables.

## Data Availability

The data presented in this study are available upon request from the corresponding author.

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
