# Peer review of "Utility of Self-Reported Heat Stress Symptoms and NGAL Biomarker to Screen for Chronic Kidney Disease of Unknown Origin (CKDu) in Sri Lanka"

_ijerph, 2021, doi:10.3390/ijerph181910498_

Round 1

Reviewer 1 Report

In the study heat stress symptoms and urine markers of chronic kidney disease were studied. It was found  that heat stress symptoms and NGAL values were higher among agricultural workers in endemic CKDu regions. The study focuses on important topic but has several limitations. The aim of the study should be better defined. Introduction could be shortened and parts could be moved to methods and discussion (lines 57 to 81). The selection criteria for study participation could be better described. It should be explained if are there any climate differences between endemic and non endemic regions. Is the temperature the same? Is there any difference in type of work between regions? Why non agricultural workers from non endemic region were not included? Why did women report higher heat stress symptoms? The characteristics of study population should be better described, there are no information about creatinine concentration, urine analysis and demographic. Are weather conditions and type of work the same in March, May and August? P value should be reported for every test. It should be clearly stated what is the most important finding of the study. 

Reviewer 2 Report

As for the experimental design, I have some thoughts on the conclusion that heat stress symptoms can be used for population screening. 

1. In Figure 3, the authors used the ROC curve to demonstrate the relationship between the heat stress symptoms (classifier) and the endemic region (outcome). Then, the authors concluded that it could be useful for population screening. I would expect the outcome was measured on the individual level rather than the population level. Could the authors elaborate on it?

Below are a few comments regarding the methods section. 

  1. In line 141, please include the version of the STATA used in the analysis and properly cite the software.
  2. In Figure 1, the label for the 2nd group, non-agriculture in the endemic region is not complete. Please re-generate the figure with smaller font size and a different angle (maybe 45 degrees to make more space for the labels). 
  3. In line 161, to 163, what are those b values? Are these coefficients? If so, could you include 95% CI as well? I will recommend presenting the regression results in a table. 
  4. In line 167, please include a p-value associated with the mean comparison (4.2 points higher)
  5. Based on Figure 4, it seems that the author might mistakenly use the variances as standard deviations of NGAL between line 186 and 228.  Please include the number of participants in the Figure 4 caption. 

Round 2

Reviewer 1 Report

Thank you for corrections. The remarks were clearly addressed. I suggest continuing your studies in this clinically significant field.